# Effects of Urbanization on the water cycle in the Shiyang River Basin: Based on stable isotope method

Rui Li[a,b], Guofeng Zhu[a,b,*], Siyu Lu[a,b], Liyuan Sang[a,b], Gaojia Meng[a,b], Longhu Chen[a,b], Yinying Jiao[a,b], Qinqin Wang[a,b]

**Affiliations:**

[a] *College of Geography and Environmental Science, Northwest Normal University, Lanzhou 730070, Gansu, China*

[b] *Shiyang River Ecological Environment Observation Station, Northwest Normal University, Lanzhou 730070, Gansu, China*

*\* Corresponding author. Email: zhugf@nwnu.edu.cn.*

**Abstract:** In water-scarce arid areas, the water cycle is affected by urban development and natural river changes, and urbanization has a profound impact on the hydrological system of the basin. Through an ecohydrological observation system established in the Shiyang River basin in the inland arid zone, we studied the impact of urbanization on the water cycle of the basin using isotope methods. The results showed that urbanization significantly changed the water cycle process in the basin, and accelerated the rainfall-runoff process due to the increase of urban land area, and the mean residence time (MRT) of river water showed a fluctuating downward trend from upstream to downstream, and was shortest in the urban area in the middle reaches, and the MRT was mainly controlled by the landscape characteristics of the basin. In addition, our study showed that river water and groundwater isotope data were progressively enriched from upstream to downstream due to the construction of metropolitan landscape dams, which exacerbated evaporative losses of river water,

and also strengthened the hydraulic connection between groundwater and river water around the city. Our findings have important implications for local water resource management and urban planning and provide important insights into the hydrologic dynamics of urban areas.

**Keywords:** Urbanization; Water cycle; Stable isotopes; River Connectivity

**1 Introduction**

According to the "2020 Global Cities Report," urban areas are currently home to more than half of the worldwide people, which amounts to 56.2%. This pattern is expected to continue over the course of the next decade, culminating in an urbanization rate of 60.4% by the year 2030. In addition, the study forecasts that by the year 2050, approximately seventy percent of the world's population would reside in urban areas (Chen et al., 2020; UN, 2019; UN-Habitat, 2020). Unlike other regions, urban regions have a substantial influence on the hydrological system, resulting in significant consequences on water balance and the water cycle (Gillefalk et al., 2021). To meet the diverse household and industrial requirements in metropolitan areas, where the population is concentrated and water demands are high, a complex interplay between natural and manmade components of the water cycle is required. These components include both natural features such as streams and groundwater, as well as human-made systems like drinking water and drainage networks (Gessner et al., 2014). Urbanization exacerbates water depletion and has far-reaching impacts on groundwater (Flörke et al., 2018; McDonough et al., 2020), affecting the environment and water availability (Bhaskar and Welty, 2015). Rapid urbanization will seriously

pressure the structure, function and water quality degradation of basin ecosystems
(Grimm et al., 2008; Sun and Lockaby, 2012; Sun et al., 2015).

Urbanization's effects on basin hydrology and the related processes have

complex and varying consequences (Caldwell et al., 2012; Martin et al., 2017). In the
past few decades, with the continuous acceleration of urbanization, human activities
in urban areas have become more frequent, and the hydrological effects of
urbanization have become more intense, attracting widespread attention worldwide
(Salvadore et al., 2015). The rise of impervious rivers in urbanized regions increases
the rate of urban water runoff, which raises the danger of urban floods (Wing et al.,
2018). In addition, high-intensity human activities have led to increased discharge of
domestic sewage and industrial wastewater, deteriorating water quality and ecological
environment (Pickett et al., 2011). Meanwhile, basin water cycle processes are
influenced by a combination of meteorological and subriver factors. It has been found
that urbanization has led to significant increases in runoff and peak flows in rivers
(Liu et al., 2018; Han et al., 2022) and has resulted in shorter runoff response times
(Anderson et al., 2022), which also exacerbates the intensity and frequency of
flooding in basins (De Niel and Willems, 2019; Blum et al., 2020). On the other hand,
the urbanization process leads to an increase in the amount of rainfall in the basin as
well as an increase in the frequency of extreme rainfall events (Shastri et al., 2015; Fu
et al., 2019; Yang et al., 2021), whereas in dryland inland river basins in arid zones
that are dependent on water resources for development, the impacts of urbanization on
the water cycle processes of the basins are still not clear, and they need to be explored
in depth the effects of urbanization on basin water cycle processes. Hence, study into
how human activities alter the features of river runoff and the water cycle within a
basin is essential for the prudent use and sustainable development of water resources.

Isotopes that are stable of hydrogen and oxygen are very useful tools for

investigating hydrological issues that are connected to river water and groundwater
sources (Fekete et al., 2006; Förstel and Hützen, 1983; Vystavna et al., 2021).
Researchers have been conducting studies using stable isotopes as tracers over the
course of the past few years in order to explore the impact that urbanization has had
on the water cycle. Urbanization has the potential to trigger and intensify convective
activity and warm-season rainfall in both urban areas and their surrounding regions
(Burian and Shepherd, 2005). Researchers generally agree that urbanization reduces
depressions on the underlying river, weakens water permeability and increases runoff.
At the same time, the lower roughness of the underlying river shortens the confluence
time (Guan et al., 2015; Oudin et al., 2018). Moreover, against the backdrop of swift
urbanization, the swift proliferation of urban regions has resulted in a sharp surge in
impermeable areas, alterations to regional microclimates, and the erection of a vast
number of infrastructures (including overpasses, subways, and so on), all of which
have significantly impacted the water cycle process in urban areas (Jacobson, 2011;
Westra et al., 2014). The complex connection between the permeable and
impermeable zones influences the river confluence processes (Bruwier et al., 2020).
The construction of urban water conservation projects, such as rubber dams and
pumping stations, also affects the confluence process of urban areas to a certain extent
(Zhu et al., 2021). Limited long-term and continuous monitoring has hampered
accurate depiction of urbanization's spatiotemporal effects on basin hydrology.
Furthermore, the scientific research till lacks sufficient research on arid regions that
heavily depend on mountain river runoff for sustenance and development.
Against the background of increasing urbanization, it is particularly important to
study the hydrological impacts of urbanization on basins and their corresponding
countermeasures, especially in arid inland river basins, where the impacts of human
activities in urban areas on rivers may be more prominent. Therefore, the Shiyang
River (SYR) basin, located in the inland arid zone of Northwest China, was used as an
example to study the impact of urbanization on the hydrology of the basin using the
stable isotope method. The following problems are proposed to be solved: (1) An
examination of the mechanisms underlying evaporation and infiltration of river water
within urban aquatic ecosystems; (2) Assessing the effects of urbanization on water
body connectivity through a comprehensive analysis; (3) The influence of
urbanization on the precipitation-runoff process is analyzed. This provides us with
essential information on how to maintain and manage the water resources found in
inland river basins, which is especially useful in light of the fact that the rate of
urbanization is growing.
**2 Observation Systems and Data**
**2.1 Study Area**
The SYR basin is located in Gansu Province, China, to the east of the He-xi
Corridor. Its coordinates are 101°41' ~ 104°16'E and 36°29' ~ 39°27'N. The SYR

basin is bounded to the west by the Wushaoling Mountain and to the north by the foothills of the Qilian Mountain (Zhu et al., 2019). The basin in question is situated within the continental temperate belt, characterized by a parched climate and diverse topography. Annual precipitation hovers within the range of 100 to 600 mm, while pan evaporation levels exhibit greater variability, ranging from 700 to 2600 mm annually. The majesty of the Qilian Mountains is where the SYR begins its journey, and the Qilian Mountains are the source of its eight main tributaries. The SYR is principally supported by the convergence of precipitation, snowmelt, and glacier runoff (Wei et al., 2013).

The Wuwei City is crossed by four important rivers, namely the Xiying, Zamu, Huangyang and Jinta, which cover a catchment area of 3986 km$^2$. As the principal water source for the entire region, the SYR basin is one of the most highly utilized inland river basins in terms of water resource development and consumption worldwide. The dams in the SYR basin are predominantly situated in close proximity to the urbanized regions of Liangzhou District, located within Wuwei City. Liangzhou District, situated in the middle of the basin, boasts of a relatively high population density and a notable commercial concentration. At the turn of the millennium, Wuwei City only boasted a paltry five landscape dams positioned on its rivers. As of 2019, this figure has surged dramatically, with a staggering total of 51 urban landscape dams now gracing both urban and peri-urban areas of the city. These dams are primarily composed of man-made landscape waterfalls and rubber dams, fulfilling their core function of creating public landscape water bodies within the urban expanse.

(Zhu et al ., 2021).

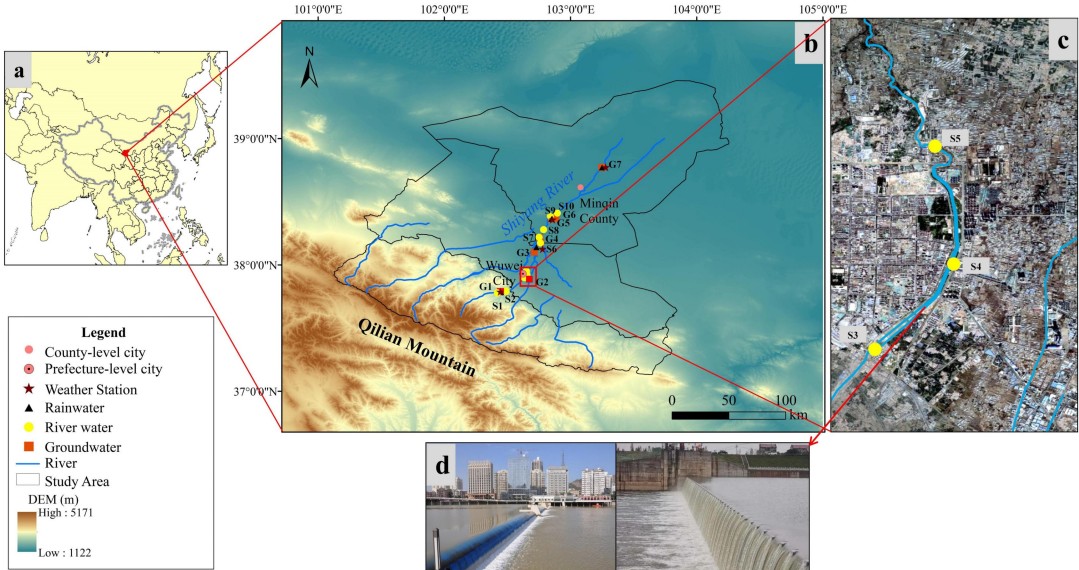

Figure 1 (a) The location of the study area, (b) Comprehensive observation system for the study
area, (c) Urban river water sampling points (from Google Maps), (d) Common urban landscape
dams in SYR basin.

## 139 **2.2 Sampling and data analysis**

Since 2017, a comprehensive observation system has been established in the
SYR basin, and stable isotope observations and hydrometeorological observations
have been carried out on river water, shallow groundwater and rainfall. Continuous
sampling in the SYR basin was carried out from April 2017 to March 2021, different
water bodies were sampled, and we collected a total of 846 samples from 24 sampling
points (Table 1). The river sampling location ought to be selected such that it is
physically possible to go as close to the middle of the river as possible, with the goal
of minimizing the impact of areas with standing water and sewage. Artesian well
water was collected as groundwater samples at 7 sampling locations around the basin.
The automated weather station was used to measure meteorological factors such as
temperature and relative humidity while collecting precipitation samples. Water
samples were sealed in high-density polyethylene bottles to avoid evaporation and
leakage during transit and storage, precipitation samples were collected using weather
station standard rain gauges. These samples were then frozen and wrapped with
plastic tape.
Table 1 Basic information on precipitation, river water and groundwater sampling sites

| Parameter | Sampling Point | Number | Sampling period | Collection Channels |
|---|---|---|---|---|
| Precipitation | P1, P2, P3, P4, P5,P6, P7, | 387 | Precipitation events | Rain tube collection |
| river Water | S1,S2,S3,S4,S5,S6, S7, S8, S9, S10 | 270 | Monthly | Sampling in river water |
| Groundwater | G1、G2、G3、G4、G5、G6、G7 | 189 | Monthly | Sampling from wells |

Analysis of the water samples is conducted through liquid water isotope analysis
utilizing the DLT-100 (Los Gatos Research, USA) in the Stable Isotope Laboratory at
Northwest Normal University. Each water sample and isotope standard are injected
six times in succession to assure reliable findings, with the first two injection values
eliminated and the average of the last four injections used for final analysis, thereby
avoiding any potential isotope analysis memory effect. The isotope measurements
were denoted by the symbol "$\delta$," which indicates the deviation in thousandths from
the Vienna Standard Mean Ocean Water:

$$\delta_{\text{sample}}(‰) = [(\frac{R_s}{R_{v-smow}}) - 1] \times 1000 \qquad (1)$$

where $R_s$ is the ratio of $^{18}O/^{16}O$ or $^2H/^1H$ in the collected sample, $Rv\text{-}smow$ is the
ratio of $^{18}O/^{16}O$ or $^{2}H/^{1}H$ of the Vienna standard sample, and the analytical accuracy
of $\delta D$ and $\delta^{18}O$ is ±0.6‰ and ±0.2‰, respectively.
**3 Methods**
**3.1 Calculation and indication of *d-excess***
Dansgaard (1964) introduced the concept of deuterium excess (*d-excess*) as the
difference in isotopic composition between global precipitation and the Vienna
Standard Mean Ocean Water ($V_{SMOW}$) reference water, which corresponds to a value of
10‰. This parameter reflects the average isotopic composition of air masses
associated with precipitation and is widely used to identify atmospheric source
regions (Deng et al., 2016). *d-excess* was proposed by Dansgaard (Dansgaard, 1964)
and is defined as:
$$d\text{-}excess = \delta D - 8\delta^{18}O \tag{2}$$
**3.2 Calculation of evaporation losses of river water**
The losses of river water through evaporation and the resulting fluctuations in
water levels of rivers, lakes, and wetlands are key aspects of the terrestrial water cycle
that merit significant attention (Gammons et al., 2006; Hamilton et al., 2005).
Evaporation is the primary mechanism of water losses in the water cycle. For river
water in dry regions and urban river water that flows slowly due to manmade
constraints, evaporation cannot be ignored. Thus, it is vital to address the alteration of
urban landscape dam water caused by non-equilibrium isotope fractionation during
evaporation. The provided formula (3) can be used to estimate the rate of evaporative
water losses from the body of water in question (Skrzypek et al., 2015):

$$f = 1 - \left[ \frac{(\delta - \delta^*)}{(\delta_0 - \delta^*)} \right]^{\frac{1}{m}}$$

(3)

The variables in the equation are as follows: $f$ represents the ratio of water lost to
evaporation, $\delta$ denotes the measured values of the water body located in the urban
dam area of Wuwei City, situated in the middle reaches of the SYR and $\delta_0$ represents
the initial value of the hydrogen and oxygen stable isotope of the water body. It is
widely assumed that the point of intersection between the local meteoric water line
(LMWL) and the local evaporation line (LEL) represents the average isotopic
composition of the input water body within the basin (Gibson et al., 2005). In the
current investigation, the intersection point marked by $\delta^{18}O$ = -7.24‰ and $\delta D$ =
-46.9‰ has been designated as the $\delta_0$ value, while $\delta^*$ denotes the maximum isotope
enrichment factor and m corresponds to the enrichment slope. The calculation of the
above parameters in this paper is realized in Hydrocalculator software (Skrzypek et al.,
2015) (http://hydrocalculator.gskrzypek.com). According to studies (Qian et al., 2007),
it is more accurate to use $\delta^{18}O$ when calculating the evaporation losses ratio, so this
study calculates the $f$ value of SYR water using $\delta^{18}O$ value.
**3.3 Periodic regression analysis and the mean residence time (MRT)**
Seasonal fluctuations in $\delta^{18}O$ values were analyzed using periodic regression
analysis to determine how these values changed over time. This method entailed
fitting seasonal sine wave curves to annual $\delta^{18}O$ variations using least squares
optimization (Rodgers et al.,2005):

$$\delta^{18}O = \delta^{18}O_{ave} + A \cdot \left[ \cos(c \cdot t - \theta) \right]$$

(4)

The modelled $\delta^{18}O$ values and the mean weighted annual measured $\delta^{18}O_{ave}$
values were both utilized in the analysis of seasonal fluctuations in $\delta^{18}O$ levels.
Additionally, the measured $\delta^{18}O$ annual amplitude ($A$), the radial frequency of annual
fluctuations ($c$), and the time in days after the start of the sampling period ($t$) were
also considered in this analysis. Furthermore, the phase lag or time of the annual peak
$\delta^{18}O$ in radians ($\theta$) was determined through this approach.
An exponential model was used for the purpose of estimating the mean residence
time (MRT). This model operates on the presumption that precipitation inputs quickly
mix with resident water. In order to do this, the following equation was used
(Maloszewski et al., 1983; Rodgers et al., 2005):

$$MRT = c^{-1} \cdot \left[ (A_{Z2} / A_{Z1})^{-2} - 1 \right]^{0.5}$$

(5)

The amplitude of precipitation ($A_{Z1}$), the amplitude of the river water outputs
($A_{Z2}$), and the radial frequency of the annual fluctuation ($c$) as defined in Eq. (4) were
taken into consideration to estimate the MRT.
**4 Results**
**4.1 Spatiotemporal distribution of isotopes in different water bodies**
The isotopes values of the river water in the SYR basin show a clear enrichment
from upstream to downstream when viewed from space. It is worth noting that
landscape dams and reservoirs in urban areas alter this pattern significantly, producing
markedly higher isotopic compositions of river water around such structures (Fig. 2).
To be more specific, the river water throughout the entire basin had average isotope
values that were lower than those of the sampling points in the dams region, which
had values that were greater (Table 2). In addition, the dams slowed the flow of the

river, this resulted in isotope enrichment of the river water. Notably, these values exhibit spatial and temporal variability, with the largest $\delta$D and $\delta^{18}$O values observed in river water, and the lowest in groundwater.

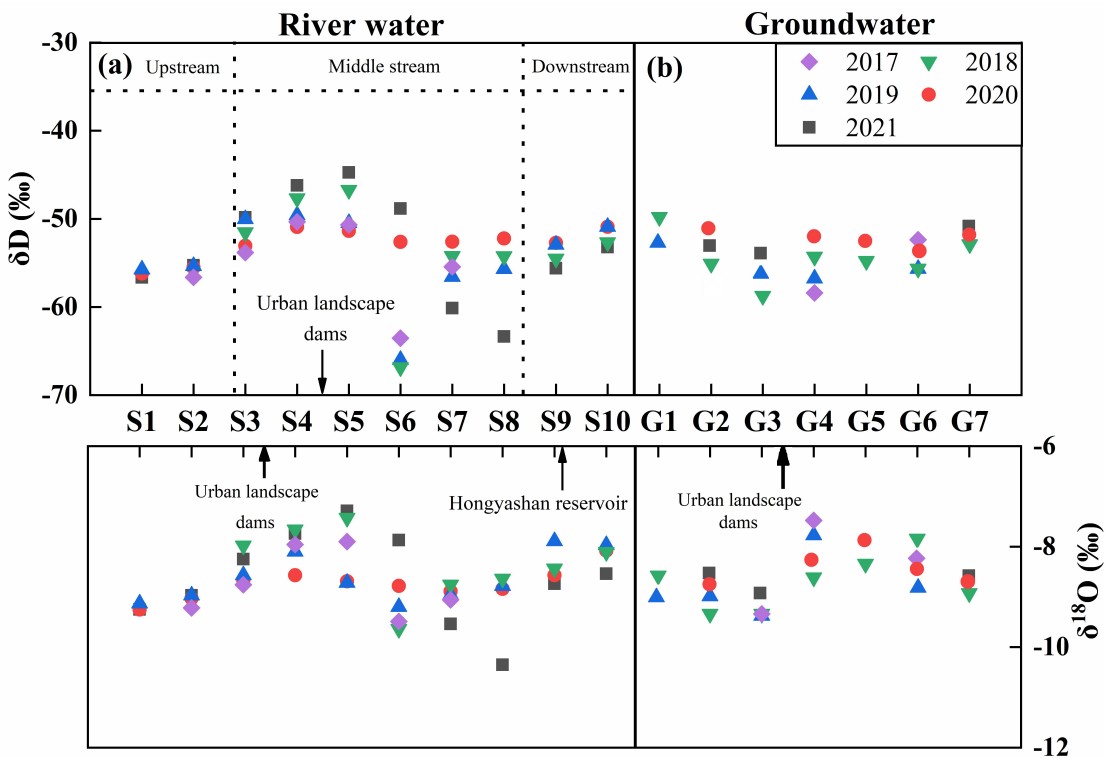

Figure 2 Longitudinal variation of $\delta$D and $\delta^{18}$O in river water and groundwater in the SYR basin.

To be more specific, over the course of time, these values shift seasonally from spring to autumn (Table 2, Fig. 3). There was a range of values from -75.43‰ to -40.62‰ for the $\delta$D values of river water, with an average of -53.53‰. The $\delta^{18}$O values display a varied range, from -10.43‰ to -5.53‰, with an average of -8.54‰, whereas the *d-excess* values demonstrate variability ranging from 10.26‰ to 29.72‰, with 15.28‰ as the average value. A broad spectrum of $\delta$D values are observed during the summer season, ranging from -61.27‰ to -31.16‰, with an average -48.90‰. Meanwhile, $\delta^{18}$O values fluctuate between -9.52‰ and -3.41‰, with an average -8.12‰. The phenomenon that was observed can be traced back primarily to

the aftereffects of the Hongyashan Reservoir built downstream. Because the reservoir

has such a large capacity for water retention, it causes significant amounts of river

water to evaporate in summer, which ultimately results in a discernible enrichment of

the isotopic composition. In both river water and groundwater, $\delta$D and $\delta^{18}$O showed

significant seasonal variations (Fig. 3). Seasonal variations were more pronounced in

river water than in groundwater, with river water showing the largest amplitude in

spring and the smallest amplitude in fall, while groundwater showed closer

amplitudes in all seasons, which also indicates that groundwater is less disturbed.

Table 2 Isotopic composition statistics of river water in SYR basin

| Sampling Point | $\delta^{18}$O | | | $\delta$D | | | d-excess | | |
|---|---|---|---|---|---|---|---|---|---|
| | Mean | Min. | Max. | Mean | Min. | Max. | Mean | Min. | Max. |
| S1 | -9.35 | -9.86 | -9.06 | -57.16 | -59.46 | -52.47 | 17.2 | 12.33 | 23.91 |
| S2 | -9.22 | -10.02 | -8.78 | -56.62 | -63.85 | -10.02 | 16.46 | 15.53 | 19.28 |
| S3 | -7.74 | -9.03 | -7.75 | -49.84 | -50.76 | -46.66 | 15.42 | 13.59 | 19.48 |
| S4 | -7.29 | -8.79 | -7.65 | -46.22 | -53.29 | -46.26 | 14.9 | 11.01 | 18.03 |
| S5 | -7.43 | -9.11 | -5.53 | -48.84 | -56.66 | -40.62 | 14.29 | 14.21 | 29.72 |
| S6 | -9.54 | -10.43 | -8.29 | -60.14 | -75.43 | -54.40 | 14.31 | 10.26 | 17.62 |
| S7 | -9.04 | -9.54 | -8.21 | -54.23 | -70.04 | -48.03 | 16.54 | 12.81 | 21.16 |
| S8 | -9.15 | -10.35 | -8.64 | -56.37 | -63.35 | -52.22 | 16.84 | 14.56 | 19.54 |
| S9 | -8.41 | -9.70 | -6.02 | -53.95 | -65.33 | -45.54 | 13.33 | 12.31 | 19.50 |
| S10 | -8.18 | -8.84 | -6.58 | -51.92 | -58.05 | -45.39 | 13.48 | 12.21 | 21.72 |

**4.2 The Relationship between $\delta$D and $\delta^{18}$O values**

As shown by the linear fitting equation $\delta$D = 7.52$\delta^{18}$O+7.58, there is a significant

linear positive correlation ($R^2$ = 0.96) between $\delta$D and $\delta^{18}$O in atmospheric

precipitation in the SYR basin (Fig. 3). It is clear that the slope (7.52) and intercept

(7.58) of the local meteoric water line (LMWL) are smaller than the global meteoric

water line (GMWL), which can be attributed to the basin's location in an inland arid

region, where precipitation disturbances are less frequent and evaporative

fractionation of precipitation is stronger. On the other hand, compared with the slopes
of the LMWL, the slopes of the river water line (RWL) and the groundwater line
(GWL) are relatively close (Fig. 3), indicating that there is a strong hydraulic
connection between groundwater and river water in the SYR basin, and the slopes of
GWL and RWL show GWL > RWL in all seasons, suggesting that the river water is
most affected by evaporation and groundwater is less affected by evaporation. In
addition, both river water and groundwater sampling points were distributed near the
LMWL, indicating that both river water and groundwater receive recharge from
precipitation. Overall, the H-O isotopic composition of river water samples from the
SYR showed a linear regression of $\delta D = 5.63\delta^{18}O - 6.11$, and the slope of RWL was
the largest in the autumn (slope = 6.65) and the smallest in the summer (slope = 5.56),
which indicated that the river water evaporated the weakest in the autumn and the
strongest in the summer.

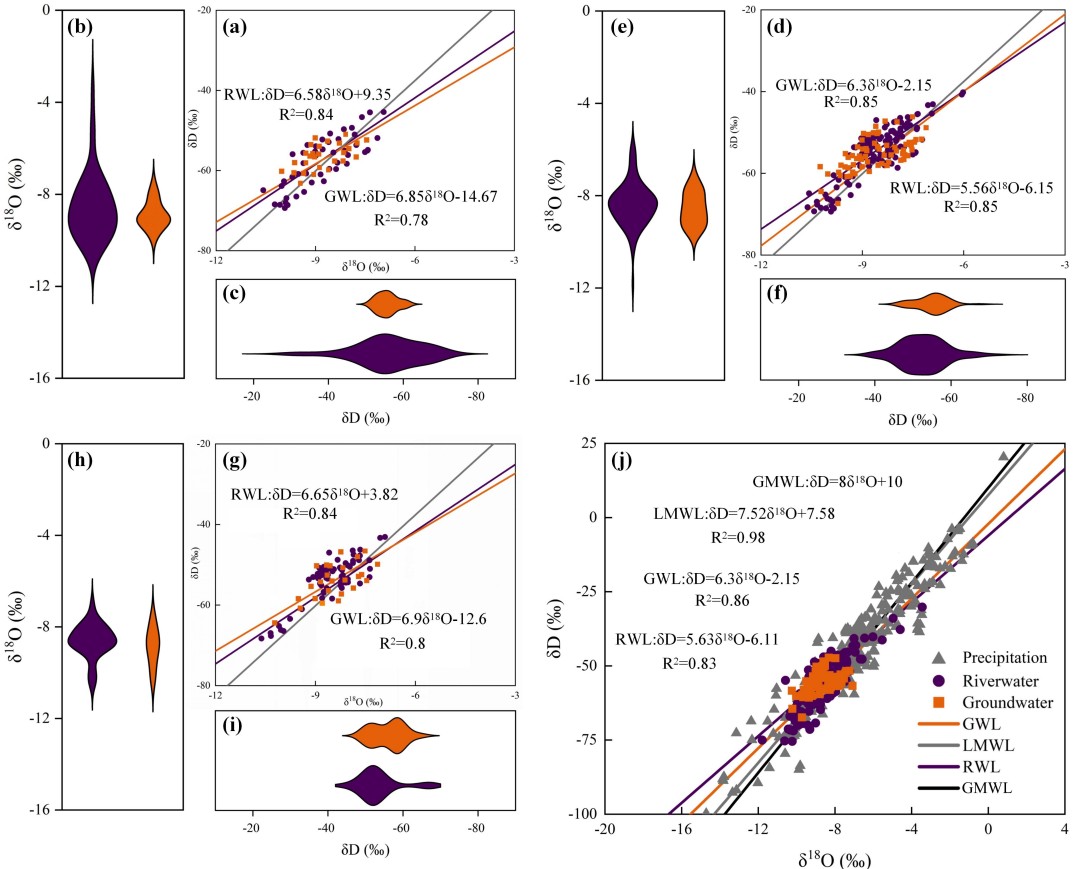

Figure 3 Relationship between $\delta$D and $\delta^{18}$O in various water bodies in the SYR basin during different seasons (a,d,g represent spring, summer, autumn; j represent the comparison of RWL, GWL, LMWL and GMWL during the entire sampling period;.b-c, e-f, h-i represent the distribution of $\delta$D and $\delta^{18}$O in river water and groundwater in spring, summer and autumn).

Isotopic analysis of groundwater samples reveals a range of $\delta$D and $\delta^{18}$O values spanning from -50.7‰ to -71.9‰ and from -7.23‰ to -10.4‰, respectively. Moreover, the groundwater samples analyzed in the study displayed a linear regression of $\delta$D=6.3$\delta^{18}$O-2.15 ($R^2$=0.86). And it is interesting to note that groundwater also shows significant enrichment near the urban landscape dams (Fig. 2), indicating that groundwater is also affected by evaporation, mainly because the Wuwei urban area is in the region of a large alluvial fan in front of the mountains, the sand and gravel aquifers are very permeable, and the depth of groundwater burial is

shallow, making the groundwater more susceptible to the effects of evaporation.
**4.3 Impact of urbanization on groundwater**
We compared monthly variations in isotopic values of groundwater near the city
with monthly variations in river water from a landscaped dam and found that the
monthly variations in groundwater near the city were closely related to river water
from a landscaped dam. The concentration of groundwater sampling sites near the city
near the sampling sites of the dam water indicates that the groundwater around the
city has similar isotopic signatures to the dam and river water (Fig. 4a). This suggests
that groundwater near the city is recharged by river water during the summer months.
In addition, we demonstrated this by comparing the data of the dam river water with
the groundwater level (Fig. 4). In addition, a portion of the groundwater sampling
sites around the city are located in the lower right corner of the LMWL, which
suggests that the groundwater around the city may also experience some degree of
evaporation.

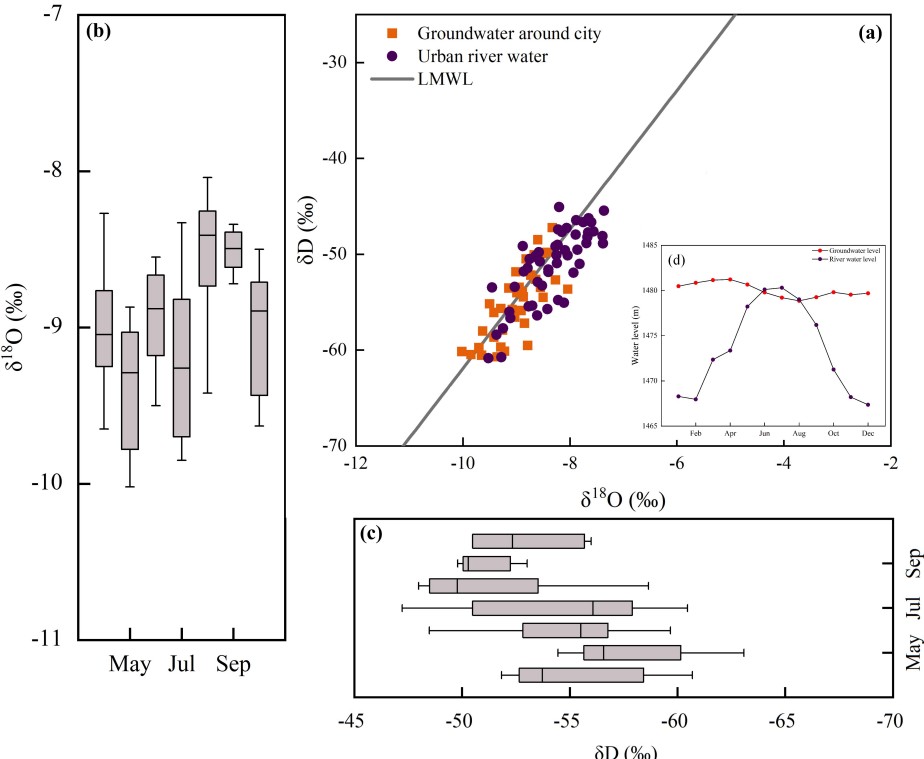

Figure 4 (a) Relationship between $\delta^{18}$O and $\delta$D of groundwater around city and urban river water; (b) Monthly variations of $\delta^{18}$O in groundwater around city; (c) Monthly variations of $\delta$D in groundwater around city; (d) Water levels in urban river water and groundwater around city.

In addition, we also compared and analyzed the changes of groundwater isotope values with those of groundwater around the city in the whole basin, and found that there was a close correlation between the changes of groundwater around the city and those of the river, while the other groundwater isotope values did not have a strong correlation with the river (Fig. 5). In the urban area, the mean values of $\delta$D and $\delta^{18}$O of the dammed river water were -7.49‰ and -48.31‰, respectively, while the mean values of $\delta$D and $\delta^{18}$O of the groundwater around the city were -8.44‰ and -50.36‰, respectively, which indicated that the $\delta$D and $\delta^{18}$O values of the groundwater around the city were similar to those of the river water in the dammed city. In addition, the isotopic mean values of $\delta$D and $\delta^{18}$O of groundwater throughout the SYR basin were

-8.73‰ and -54.78‰, which are significantly different from the isotopic values of
river water in the urban dam.

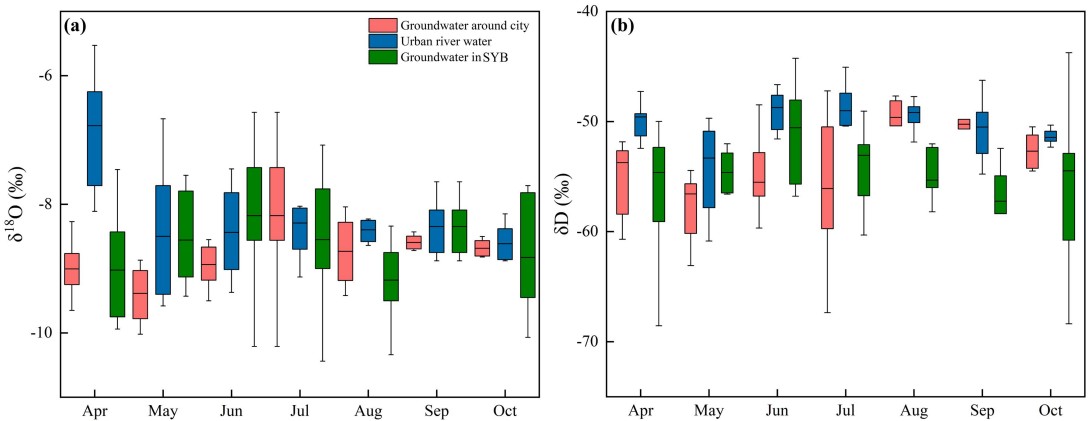

Figure 5 (a) Monthly variations of $\delta^{18}$O in urban river water and groundwater around city, (b)
Monthly variations of $\delta$D in urban river water and groundwater around city.

## 4.4 Temporal and spatial variation of river water evaporation losses in the urban area of Wuwei

In addition to being an essential part of the hydrological cycle, evaporation is
widely recognized as one of the most significant factors driving climate change in
semi-arid regions and in telluric ecosystems (Gibson et al., 2002; Gibson and Edwards,
2002). An obviously spatial and temporal fluctuation can be seen in the amount of
river water that is lost to evaporation in the upper mountain area as well as the
intermediate urban area of the SYR basin (Fig. 6). Analyzed from a time-varying
perspective, there is significant seasonal variation in river water evaporation losses
both in the upstream mountainous region and the midstream urban area of Wuwei,
with the highest rates occurring during summer and the lowest during winter (Fig.6).
Additionally, a spatial comparison reveals that river water evaporation losses in the
midstream urban area of Wuwei are significantly greater than those in the upstream
mountainous area.

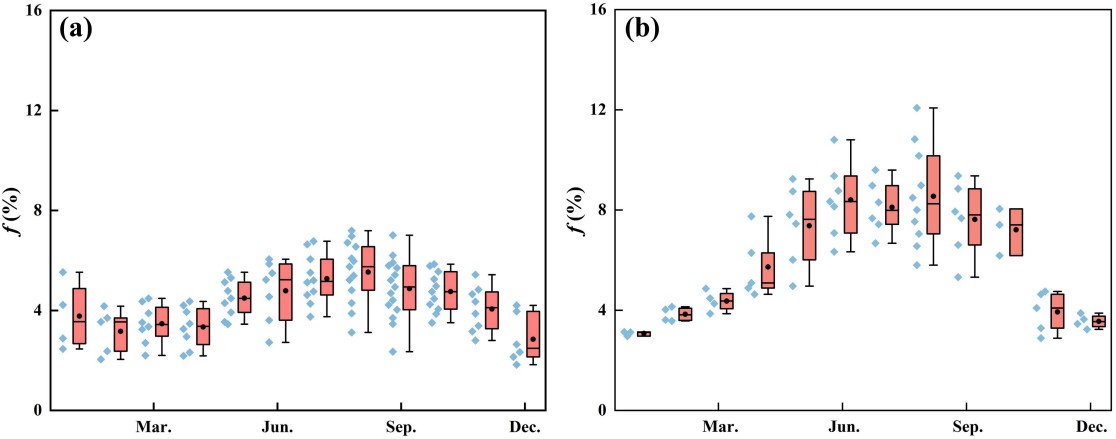


Figure 6 Evaporation losses from river water in different areas of the SYR (a) Upper reaches
mountainous area, (b) Middle reaches urban areas.
Differences contributing to evaporation losses from the river in the upstream and
midstream urban areas can be explained mainly by the landscape characteristics of the
basin. In the upstream of the SYR higher vegetation cover and atmospheric humidity
in the mountainous areas result in weaker evaporation losses, while the midstream are
dominated by urban land, and urban landscapes increase the watershed area and slow
down the river, exacerbating evaporation losses from the river.
**5 Discussion**
**5.1 Effects of Urbanization on the Rainfall-Runoff Process**
Fig. 7 depicts the regression model of rainfall events in the SYR basin,
represented by a sine wave, and the fitting of river water $\delta^{18}O$ across the research
season. The $\delta^{18}O$ levels of precipitation reported in the SYR basin have an excellent
regularity ($R^2$=0.46) and a seasonal patterns trend that effectively depicts the nfluence
of the monsoon climate on the local environment (Zhu et al., 2019). Seasonal
variations are seen in the generally steady $\delta^{18}O$ and $\delta^{18}O$ values of the upstream water.
These results indicate that the predominant component of the river water is the
baseflow resulting from recent precipitation runoff. Throughout the duration of the
study, the majority of the lowest $\delta^{18}O$ values in the 10 river water sample points were
recorded during the winter, whilst the highest values were recorded during the
summer. These trends coincide with both the temporal variation of precipitation
isotopes in the SYR basin, indicating that precipitation input is the underlying cause
of isotope changes in river water. Nevertheless, variations in the isotopes of river
water differ in range across various regions within the SYR basin, with significant
variation in the degree of fit for the regression curve. The fitting degree of river water
in the upper and lower reaches is relatively low ($R^2$=0.37, $R^2$=0.28, $R^2$=0.23),
implying limited seasonal isotopic variability in these regions. The midstream river
water exhibits a notably higher degree of conformity as compared to its upstream and
downstream counterparts ($R^2$=0.38, $R^2$=0.48, $R^2$=0.62, $R^2$=0.78, $R^2$=0.54, $R^2$=0.48,
$R^2$=0.52). Moreover, the isotopic composition of river water throughout this area
exhibits notable cyclic variations.

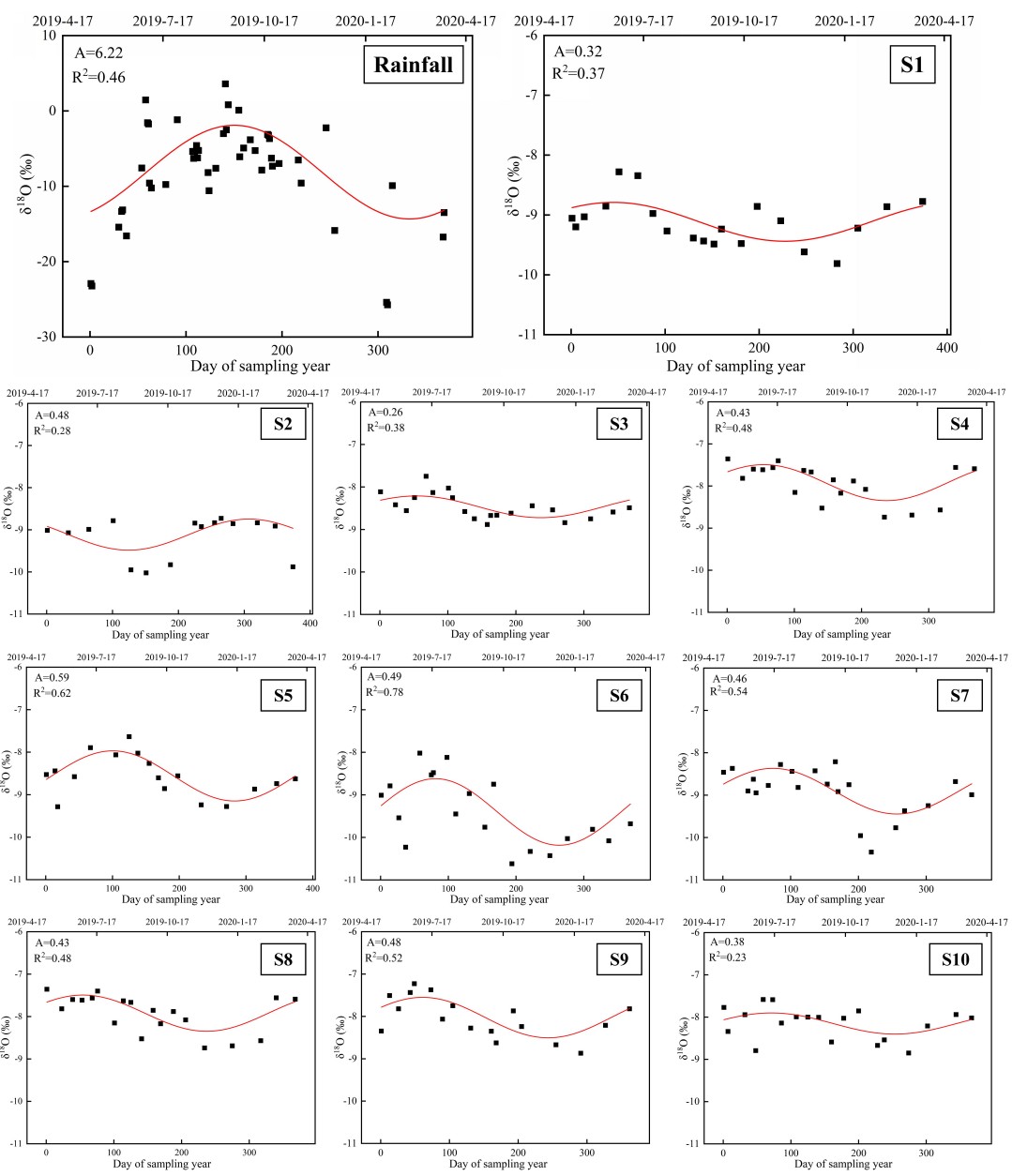

Figure 7 Fits the annual regression model of $\delta^{18}O$ in SYR basin precipitation and river water (time: 2019/4/17—2020/4/23; S1-S10 are river water sampling points).

The reasons for differences in isotope periodicity in different regions may be attributed to local water management systems, topographic features and urban development. At points S1, S2, and S10, the correlation of model simulations was low, which could be attributed to the presence of Xiying Reservoir in the upstream as well as Hongyashan Reservoir in the downstream (Sang et al., 2023), where seasonal

variations in the isotope values of the river water are interfered by the reservoir
dispatching activities. At points S3 to S5, the correlation of the model simulation is
higher, which is because in the middle reaches of the SYR basin, the expansion of
urban built-up areas leads to a significant increase in river runoff during the rainy
season, and according to the land use data, the land area of the towns in Wuwei City
has continued to increase by 134.38 km$^2$ from 2010 to 2018, resulting in the river
water showing a cyclical trend comparable to that of the precipitation. Since the 1950s,
in order to better utilize water resources, 13 small and medium-sized reservoirs with a
total storage capacity of 900,000 m$^3$ were constructed during this period (Ma et al.,
2010), increasing the proportion of rainfall in the runoff constituents as a result of The
correlation of the model simulation is at a high level at points S6~S9, where, in
contrast to the high-elevation areas in the upper reaches, the terrain in the middle and
lower reaches of the SYR basin is relatively flat, mainly with cultivated land and
deserts, and is less disturbed by human activities (Sun et al., 2021), which further
reflects the responsiveness to recent precipitation inputs.
The Dunnett's test revealed a significant difference (P < 0.05) between the MRT
of the river and the annual magnitude of $\delta^{18}O$ of the river. We further investigated the
relationship between the estimated mean residence time and basin landscape features
such as topography (Fig. 8). Using the digital elevation model (DEM) to calculate the
mean slope of the SYR basin, we found that the mean residence time was also
strongly correlated with the mean basin slope (R$^2$ = 0.63), and that the upper reaches
of the SYR basin are mainly high-elevation mountainous areas, where the topography
is sloped, but where the vegetation cover is high and dominated by alpine meadows,
subalpine scrub and Qinghai spruce (Zhang et al. 2023), the greater slope leads to a
higher gravitational potential, which tends to result in a negative correlation with
mean residence time (McGuire et al., 2005), which also contributes to the potentially
higher MRT values in the upstream mountains. In our study, catchment area (CA) had
a low correlation with MRT ($R^2 = 0.40$), and a weak relationship between catchment
area and MRT has been observed in other studies (McGlynn et al., 2003; McGuire et
al., 2005).

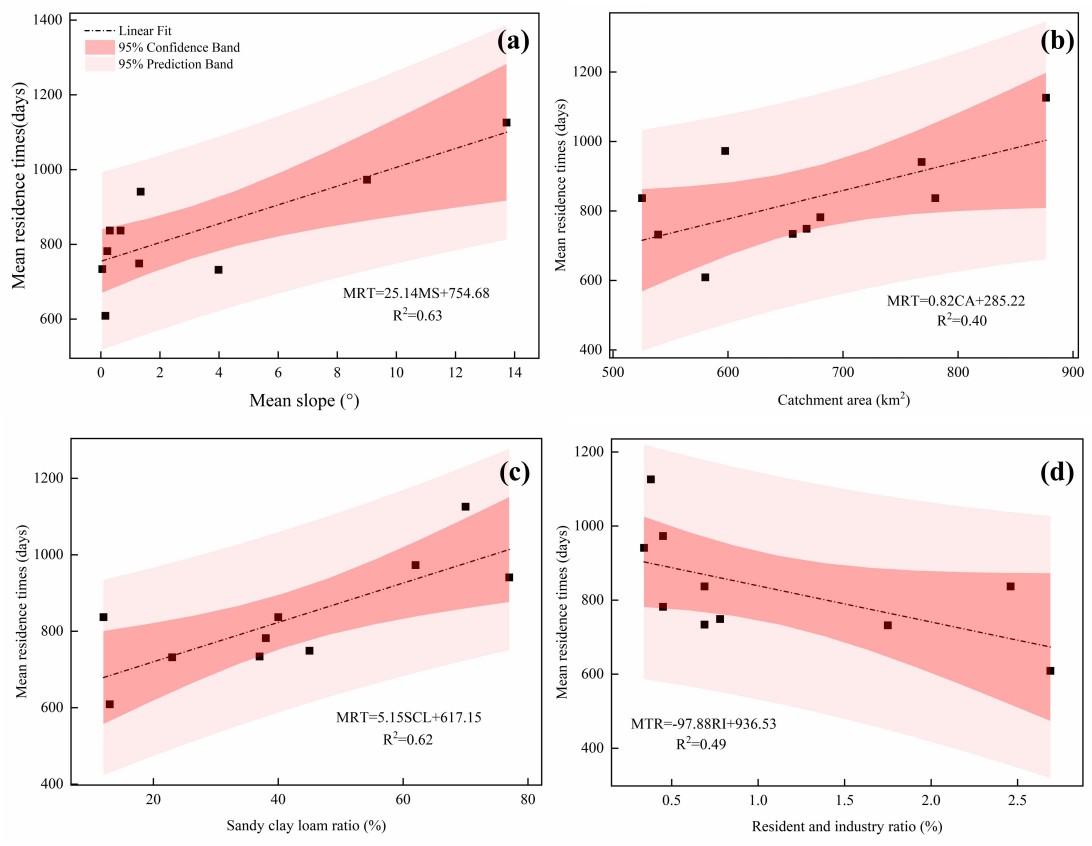

Figure 8 Correlation between MRT and (a) mean slope of the basin, (b) catchment area, (c) sand
clay loam ratio, and (d) percentage area of residential and industrial use in the basin, with 95%
Confidence and Prediction bands.
Soil is an important component of basin hydrology, and the physical properties

of soil, such as water-holding capacity and pore space distribution, have an important

influence on the response to precipitation in the basin and the sand-clay-loam soil

ratio is used here to investigate the possible relationship with MRT. The results

showed that the content of sand clay loam ratio showed a strong positive correlation

with MRT ($R^2$=0.62). Wuwei City is located in the pre-mountain flood-fan belt, and

the soil is dominated by sandy soil (Zhang et al., 2023), which is loose in texture, has

good permeability and good water retention properties, and is mainly used for

agricultural cultivation. Its good permeability increases the vertical movement of

water and the length of flow paths, leading to a longer MRT. There is a strong

negative correlation between the MRT and the ratios of resident and industrial areas

(RI) ($R^2$=0.49), which also indicates that as urbanization progresses, with the increase

of urban land, this undoubtedly leads to a significant shortening of the MRT. However,

the MRT in the mid-river urban area is not much shorter as compared to the

downstream, which may be attributed to the fact that the mid-river The large number

of landscape dams constructed in the urban areas, currently 51 urban landscape dams

have been built in the peri-urban areas of Wuwei City, and the considerable number of

landscape dams may have counteracted the impact of the urban land use, resulting in a

lengthening of the MRT in the middle reaches as well.

**5.2 Effects of Water Conservancy Projects in Urban Areas on Isotope Dynamics**

Recent studies have suggested that the development of dam-reservoir systems

may result in river fragmentation and modifications in flow regimes in terms of their

volume,frequency,and duration (Négrel et al., 2016; Murgulet et al., 2016; Peñas and

Barquín, 2019; Maavara et al., 2020). Furthermore, chemical-containing nutrient migration, such as phosphorus, may occur during sediment movement, resulting in widespread eutrophication problems (Yang et al., 2007; Duan et al., 2019). As of 2019, a total of 51 urban landscape dams, primarily consisting of artificial landscape waterfalls and rubber dams, have been constructed in and around Wuwei city (Zhu et al., 2021). In the metropolitan coast of Wuwei, many landscape dams have led to isotopic enrichment in river water. This damping effect has been observed in numerous dammed rivers across the globe, including the Rio Grande in the southwestern United States (Vitvar et al., 2007) and the Ebro River in Spain (Négrel et al., 2016), as evidenced by isotopic tracers. In the metropolitan coast of Wuwei, a number of landscape dams have led to the enrichment of isotopic tracers in the river water. The results indicate that the $\delta$D and $\delta^{18}$O levels of the river water at the outflow of Wuwei City are greater than those at the inflow (Fig. 2). Moreover, the influence of evaporation on isotopic composition should not be overlooked, as it can lead to a decrease in *d-excess* values (Peng et al., 2012). Consistent with previous studies (Wang et al., 2019), we observed that the *d-excess* of influent water was higher than that of urban river water (Fig. 9). This observation further supports the accumulation of heavy H-O isotopes in the river waters of the dam areas, as shown in Fig. 9a. In contrast, due to the confluence of tributaries prior to the S7 sampling point, the river water has lower isotopic values, resulting in elevated *d-excess* values between S6 and S8.

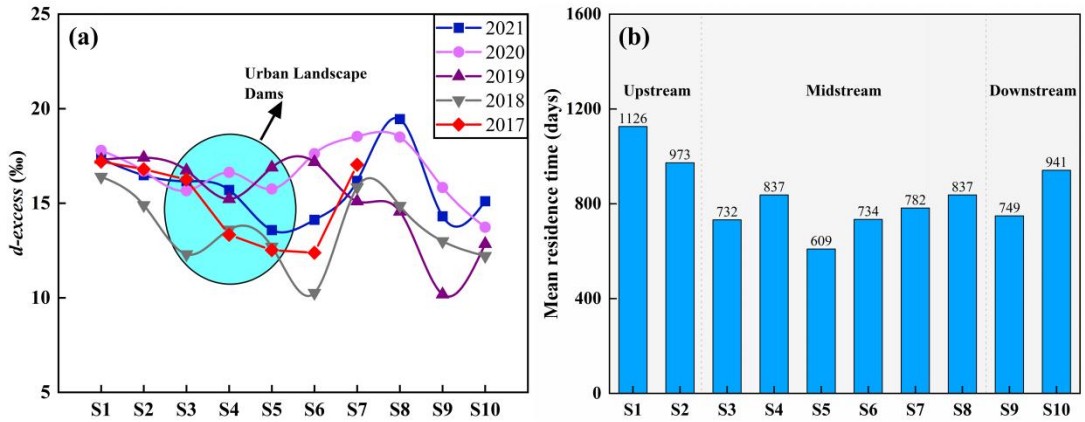

Figure 9 (a) The longitudinal variation of the river water *d-excess* of the SYR, (b) The longitudinal

variation of the river water MRT of the SYR.

**5.3 Effects of Urbanization on the Water Cycle of basins**

Localized microclimates in urban areas allow for changes in precipitation and

evapotranspiration processes, while urbanization alters the pristine subriver,

complicating water cycle processes in the basin (Jacobson, 2011; Westra et al., 2014;

Oudin et al., 2018). In terms of the impact on runoff, it is mainly reflected in the

increase of river impermeability due to urbanization, the land use area of Wuwei

urban land increased by about 134.38 km$^2$ from 2010 to 2018, which greatly

weakened the infiltration process in urban areas, and the rainfall runoff process

simulated by sinusoidal cyclic regression method showed that there were significant

differences in the river metro in different parts of the SYR basin, and that the middle

reaches of the river had the highest degree of urbanization, and the time of the metro

was the shortest, which further increases the contribution of rainfall to runoff.

Regarding the effect of urbanization on evapotranspiration, a large number of dams

were constructed on the SYR and flowed through the urban area of Wuwei, causing

significant evapotranspiration losses, in addition, these landscape dams also led to

hydrogen and oxygen isotope enrichment, and the numerous reservoirs that were
constructed for the construction and development of the city (Ma et al., 2010), and
these reservoirs also contributed to a significant evapotranspiration loss effect, which
has been previously confirmed in our study was also confirmed (Sang et al., 2023).
On the other hand, our study found that the isotopic compositions and trends of urban
nearshore groundwater were similar to those of river water, which suggests that there
is a close correlation between urban nearshore groundwater and river water, and that
the difference in water levels between river water and groundwater may be the main
reason for river recharge of urban nearshore groundwater (Fig. 4d). In the rainy
season, the river level gradually rises, which decreases the difference between the
water levels of urban nearshore groundwater and river water, and the river water
recharges the groundwater, and in the dry season, the river level decreases, and the
urban nearshore groundwater, which is buried at a shallow depth, in turn recharges the
river.
In addition, the growth of urbanization has had a dramatic impact on the water
environment in cities, where water problems occur frequently (Giri and Qiu, 2016;
Ma et al., 2022). Urbanization has increased impervious rivers such as parking lots,
rooftops, roads, and sidewalks, leading to increased runoff, which creates additional
pathways for pollutants to be transported from landscapes to water bodies (Ren et al.,
2014; Wilson and Weng, 2010; Nolan et al., 2023). On the other hand, agricultural
activities have increased some of the fertilizers, pesticides, herbicides and dairy
manure in the farmland into the nearest water bodies, which can directly and
indirectly affect will reduce water quality (Yu et al., 2013). The SYR basin in the
Northwest Arid Zone is an inland river basin with the highest development intensity
and the sharpest conflict between water supply and demand in the region. The
Liangzhou district in the central part of the SYR basin is the most densely populated
artificial oasis with the largest scale of water demand in the entire basin. Our previous
study found that direct discharge of industrial and community domestic wastewater
into the river led to deterioration of river water quality around the SYR basin (Ma et
al., 2021). In addition agricultural activities have less impact on the upper reaches of
the SYR and relatively more impact on the middle and lower reaches , and the
application of nitrogen-based fertilizers during agricultural cultivation is the main
cause of high $NH4^+$ and $NO3^-$ concentrations in the area (Ma et al., 2021), which may
also lead to increased salinity and accelerated eutrophication of the river, threatening
the safety of the basin's water environment. Overall, human activities (urbanization)
may alter the water cycle processes inherent in inland river basins, and the
implications of such changes need to be further explored.
**6 Conclusions**
In this study, we investigated the hydrometeorological and isotopic data of the
Shiyang River basin from 2017 to 2021, and our investigations showed that
urbanization had a significant impact on the water cycle of the basin.The results
showed that the isotopic values of the river water showed a significant enrichment
from upstream to downstream, but facilities such as landscape dams and reservoirs in
the urban area significantly altered this natural pattern, and the isotopic values of the

river water in the urban area ($\delta$D=-48.31‰; $\delta^{18}$O=-7.49‰) were higher than those of

the natural river water ($\delta$D=-55.77‰; $\delta^{18}$O=-8.98‰), and landscape dams aggravated

the evaporation losses of river water, due to the increase of urban land area, which

accelerated the rainfall-runoff conversion process, the residence time of river water in

different regions of the Shiyang River basin had obvious differences, and the MRT

from the upstream to the downstream showed a fluctuating downward process, which

was shortened from 1,126 days in the upstream to 941 days in the downstream, and

the MRT was mainly controlled by the basin's landscape features. In addition, there

was a strong relationship between the isotopic composition of the reservoir and the

surrounding groundwater. Overall, urbanization has a profound impact on the

hydrological system of the basin, and the results of this study can provide some

references for future research on urbanization and the water cycle, and improve our

understanding of the hydrological processes of basin in arid zones.

**Acknowledgements**

This research was financially supported by the National Natural Science

Foundation of China (41971036, 41867030).

**Author contributions statement**

Rui Li: Writing-Original draft preparation; Guofeng Zhu: Writing-Reviewing and

Editing; Siyu Lu: Methodology; Liyuan Sang and Gaojia Meng: Data processing and

Experiment; Longhu Chen and Yinying Jiao: Methodology and visualization;

Qinqin Wang : Visualization;

**Data availability Statement**

The isotopic data that support the findings of this study are openly available in Zhu, Guofeng (2022), "Stable water isotope monitoring network of different water bodies in SYR Basin, a typical arid river in China", Mendeley Data, V1, doi: 10.17632/vhm44t74sy.1. The source of soil data comes from the Harmonized World Soil Database (HWSD) constructed by the Food and Agriculture Organization of the United Nations (FAO) and the International Institute for Applied Systems (IIASA) on 2009. The land-use and land-cover change data of the Shiyang River Basin were obtained from Chinese Academy of Sciences, the data centre of resources and environmental science (http://www.resdc.cn).

**Competing Interests**

We undersigned declare that this manuscript entitled "Effects of Urbanization on the water cycle in the SYR Basin: Based on stable isotope method" is original, has not been published before and is not currently being considered for publication elsewhere.

The authors declare that they have no known competing financial interests or personal relationships that could have appeared to influence the work reported in this paper.

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
