# Peer review of "Effects of Urbanization on the water cycle in the Shiyang River"

_Hydrology and Earth System Sciences, 2023_

## Author Comment (AC1)

**Manuscript Number: hess-2023-96**

**Revision notes:**

We would like to thank the editor and reviewers for the valuable and comments, which were very helpful for the further improvement of the manuscript. We have looked through each comment the editor and reviewers raised and responded and incorporate it in the revised manuscript. The following is the detailed response:

**Reviewer #3:**

The authors presented a case study to infer the impact of urbanization on water cycle in the Shiyang River Basin using isotope data. With the support of large datasets of stable isotopes for a whole river catchment, the manuscript has clarified the urban landscape dams have a significant impact on the water cycle. As my assessment, these datasets are really valuable and innovative. I suggest an acceptance after moderate revisions. Some obvious shortcomings are as following points but not limited these as listed. There are so many unexpected errors included in the whole text.

**Response:** Thank you for providing your valuable comments and suggestions. Based on your comments and suggestions, we have thoroughly revised the manuscript in accordance with the comments you have made, and the revisions are indicated in blue.

**General comments:**

1. The authors collected a large number of samples, such as precipitation, surface water and groundwater. However, the paper mainly analyzed the surface water, and almost did not analyze the groundwater. The paper should strengthen the study of groundwater in the hydrological cycle.

**Response:** In response to your suggestion, we have strengthened our study of groundwater in the hydrologic cycle and analyzed the possible impacts of urbanization on groundwater, and the following is the detailed information on our modifications.

Specifically:

We compared the isotope values of groundwater near the city with the monthly variations of river water from landscaped dams, and found that the monthly variations of groundwater near the city were closely related to the river water from landscaped dams. The concentration of groundwater sampling sites near the city near the sampling sites of the dam water indicates that the groundwater around the city has similar isotopic signatures to the dam and river water. This suggests that groundwater near the city is recharged by river water during the summer months. In addition, we demonstrated this by comparing the data of the dam river water with the groundwater level. In addition, a portion of the groundwater sampling sites around the city are located in the lower right corner of the LMWL, which suggests that the groundwater around the city may also experience some degree of evaporation.

[Figure]

Figure 4 (a) Relationship between $\delta^{18}O$ and $\delta D$ of groundwater around city and urban river water; (b) Monthly variations of $\delta^{18}O$ in groundwater around city; (c) Monthly variations of $\delta D$ in groundwater around city.

In addition, we also compared and analyzed the changes of groundwater isotope values in the whole basin with those around the city, and found that there was a close correlation between the changes of groundwater around the city and those of the river, while the other groundwater isotope values did not have a strong correlation with the river. In the urban area, the mean values of δD and δ¹⁸O of the dammed river water were -8.26‰ and -49.88‰, respectively, while the mean values of δD and δ¹⁸O of the groundwater around the city were -8.44‰ and -50.36‰, respectively, which indicated that the δD and δ¹⁸O values of the groundwater around the city were similar to those of the river water in the dammed city. In addition, the isotopic mean values of δD and δ¹⁸O of groundwater throughout the SYR basin were -8.73‰ and -54.78‰, which are significantly different from the isotopic values of river water in the urban dam.

[Figure]

Figure 5 (a) Monthly variations of δ¹⁸O in urban river water and groundwater around city, (b) Monthly variations of δD in urban river water and groundwater around city.

2. Urbanization has many influences on the water cycle, such as urban sewage and landscape dam, etc. This paper mainly analyzes the influence of landscape dam elements, while other possible influences on the water cycle are only hypothesized and lack of data support. What is described in the article is the impact of urbanization, rather than the impact of landscape dams on the water cycle.

**Response:** We fully agree with your comments, As you pointed out, the impact of urbanization on the water cycle is multifaceted, here we selected the Shiyang River Basin in the arid region of Northwest China to explore the impact of urbanization on the local water cycle, it is worth noting that water resources is one of the most important factors limiting the development of the local area, therefore, we selected the example of urban landscape dams in the process of urbanization to analyze the impact of the urban water conservancy facilities on the water cycle, because these dams cause a large amount of evaporation loss, which has an impact on the utilization of water resources. On the other hand, we also studied the runoff process of the water cycle and the interaction between groundwater and surface water. The results of the study showed that the increase in urban land use due to urbanization exacerbated the process of rainfall-runoff transformation, which led to a significant shortening of the residence time of surface water, and that groundwater near the urban area was recharged by the river water flowing through the landscape dams in the summer season, and that, due to the shallow depth of the groundwater buried near the city , groundwater is affected by evapotranspiration.

3. In the Introduction section, a large part describes the impact of urbanization on the water cycle of urban areas. However, the article mainly focuses on river basins. What is the current research on the impact of urbanization on the water cycle in river basins.

**Response:** According to your suggestion, we have revised the introduction section to strengthen the description of urbanization on the water cycle of the basin, and the detailed revision information is as follows.

Specifically:

Meanwhile, basin water cycle processes are influenced by a combination of meteorological and subsurface factors. It has been found that urbanization has led to significant increases in runoff and peak flows in rivers (Liu et al., 2018; Han et al., 2022) and has resulted in shorter runoff response times (Anderson et al., 2022), which also exacerbates the intensity and frequency of flooding in basins (De Niel and Willems, 2019; Blum et al., 2020). On the other hand, the urbanization process leads to an increase in the amount of rainfall in the basin as well as an increase in the frequency of extreme rainfall events (Shastri et al., 2015; Fu et al., 2019; Yang et al., 2021), whereas in dryland inland river basins in arid zones that are dependent on water resources for development, the impacts of urbanization on the water cycle processes of the basins are still not clear, and they need to be explored in depth the effects of urbanization on basin water cycle processes.

Response: Due to our negligence, which caused this error, we have modified Figure 4, It is now named Figure 6, and the modified picture is as follows.

[Figure]

Figure 6 Evaporation losses from surface water in different areas of the SYR (a) Upper reaches mountainous area, (b)Middle reaches urban areas.

3. Line 309-310: The meaning of the sentence is unclear

**Response:** I'm sorry that this sentence was not clearly expressed, which has caused you a misunderstanding. We have amended the formulation of this sentence. The detailed explanation is as follows.

We have changed "Seasonal variations are seen in the generally steady $\delta^{18}O$ and $\delta^{18}O$ values of the upstream water." to "Runoff formed by upstream river water with relatively stable $\delta^{18}O$ and rainfall with significant seasonal variations in $\delta^{18}O$ are the two main components of river water."

4. Line 311-312: Analysis of precipitation and surface water alone is not enough to support conclusions, such as whether groundwater or urban human water use has the same effect.

**Response:** We fully agree with your comments, the stability of the groudwater isotope composition over time has been demonstrated in Figs. 2 and 3, so we assume that the surface runoff in the urban area mainly originates from the contribution of precipitation, and the main focus of this study is the surface runoff in the urban area, so it is assumed that the groundwater does not contribute to the runoff. In addition, we calculated the sinusoidal regression cycles of precipitation and river water in the SYR basin separately, in order to reveal the cyclic changes of runoff and precipitation in urban areas, and to reveal the influence of urbanization on the runoff process in the basin.

5. Part 5.1: The authors collected 3 years of data, but only used 1 year of data as a simulation, the validity of this year's data?

**Response:** We have explained the key issue you mentioned. The detailed explanation is as follows.

We have set up a monitoring system in the Shiyang River Basin since 2017, but due to the influence of sampling conditions and other factors, the precipitation data in 2017 and 2018 are missing, and the time series is not continuous. The data integrity of the precipitation sampling point we set up in 2019 and 2020 is the highest. Therefore, we choose the precipitation data from 2019 to 2020 to do the periodic regression analysis of precipitation and surface water.

6. Figure 6: "landscape" not "landsacpe"

**Response:** I am very sorry for our negligence. We have modified the Figure 6, It is now named Figure 9, and the specific modification is as follows.

[revised manuscript text omitted]

---

## Author Comment (AC2)

**Manuscript Number: hess-2023-96**

**Revision notes:**

We would like to thank the editor and reviewers for the valuable and comments, which were very helpful for the further improvement of the manuscript. We have looked through each comment the editor and reviewers raised and responded and incorporate it in the revised manuscript. The following is the detailed response:

**Reviewer #2:**

hess-2023-96 "Effects of urbanization on the water cycle in the Shiyang River Basin: Based on stable isotope method" Rui Li, Guofeng Zhu, Siyu Lu, Liyuan Sang, Gaojia Meng, Longhu Chen, Yinying Jiao, Qinqin Wang

This article is a presentation of a reasonable data collection exercise but with no unique or local conditions that distinguish it from published literature. This is a disappointing article which seems to boil down to "urban areas with dams cause water to slow allowing enhanced evaporation". In the Discussion, mentions of the water cycle are all speculative with no direct support from the data collected.

The water cycle didn't really get investigated.

**Response:** Thank you for providing valuable comments and suggestions. The suggestions provided were carefully incorporated into the revised manuscript. We have gradually established a complete ecohydrological observation system in the Shiyang River Basin since 2017, and have analyzed the mechanism of urbanization's impact on the water cycle process in the basin based on the meteorological and hydrological data we have collected. We have again analyzed the collected isotope data, enhanced the analysis of the groundwater section, strengthened the argumentative analysis of the results and discussion sections of the manuscript, and presented our findings with more graphs. All of your comments and suggestions have helped us improve the revised manuscript, with the revised sections in blue. In addition, for your review, we have included a PDF of the revised manuscript at the end of this document.

1. The groundwater data was just a sideline that never got explored or tied to anything else.

**Response:** In response to your suggestion, we have strengthened our research on groundwater in the hydrological cycle and have studied the impact of urbanization on groundwater, and the following is the detailed information on our modifications. Specifically:

We compared the monthly changes in isotope values of groundwater near the city with the monthly changes in isotope values of river water from a landscaped dam in the city, and found that the monthly changes in groundwater near the city were closely related to the river water from the landscaped dam (Fig. 4). The concentration of groundwater sampling sites near the city near the sampling sites of the dam water indicates that the isotopic signature of groundwater around the city is similar to that of the dam and river water. This suggests that groundwater near the city is recharged by river water during the summer months. In addition, we demonstrated this by comparing the data of the dam river water with the groundwater level. In addition, a portion of the groundwater sampling sites around the city are located in the lower right corner of the LMWL, which suggests that the groundwater around the city also experiences some degree of evaporation.

[Figure]

Figure 4 (a) Relationship between $\delta^{18}O$ and $\delta D$ of groundwater around city and urban river water; (b) Monthly variations of $\delta^{18}O$ in groundwater around city; (c) Monthly variations of $\delta D$ in groundwater around city.

In addition, we also compared and analyzed the changes of groundwater isotope values in the whole basin with those around the city, and found that there was a close correlation between the changes of groundwater around the city and those of the river, while the other groundwater isotope values did not have a strong correlation with the river (Fig. 5). In the urban area, the mean values of $\delta D$ and $\delta^{18}O$ of the dammed river water were -8.26‰ and -49.88‰, respectively, while the mean values of $\delta D$ and $\delta^{18}O$ of the groundwater around the city were -8.44‰ and -50.36‰, respectively, which indicated that the $\delta D$ and $\delta^{18}O$ values of the groundwater around the city were similar to those of the river water in the dammed city. In addition, the isotopic mean values of $\delta D$ and $\delta^{18}O$ of groundwater throughout the SYR basin were -8.73‰ and -54.78‰, which are significantly different from the isotopic values of river water in the urban dam.

[Figure]

Figure 5 (a) Monthly variations of $\delta^{18}O$ in urban river water and groundwater around city, (b) Monthly variations of $\delta D$ in urban river water and groundwater around city.

2. The rainwater data was compared to GMWL but not tied closely to other components.

**Response:** Based on your suggestions, we have enhanced our analysis and discussion of the precipitation data, and detailed information on the changes is provided below. Specifically:

The local atmospheric water line equation for the Shiyang River basin obtained from the least-squares fitting is $\delta D=7.82\delta^{18}O+7.58$, and both the slope and intercept are smaller than that of the global atmospheric water line (GMWL:$\delta D=8\delta^{18}O+10$), which indicates that the climate in the inland northwest is dry and evaporation is strong. On the other hand, compared with the slopes of the atmospheric water line, the slopes of the surface water line (SWL) and the groundwater line (GWL) are relatively close (Fig. 3), indicating that there is a strong hydraulic connection between groundwater and river water in the SYR basin, and the slopes of GWL and RWL show GWL > RWL in all seasons, suggesting that the river water is most affected by evaporation and groundwater is less affected by evaporation. In addition, both surface water and groundwater sampling points were distributed near the LMWL, indicating that both river water and groundwater receive recharge from precipitation.

3. Surface runoff was all described textually with no numbers so any concept of dilution or concentration, or changed runoff generation behaviour, was just references to other work.

**Response:** We observed that the groundwater level in the urban areas of the Shiyang River Basin is generally greater than 10 m, and there is basically no possibility of groundwater recharge runoff. We calculated the sinusoidal regression cycles of $\delta^{18}O$ for precipitation and river water in the SYR basin separately, in order to reveal the cyclical changes in $\delta^{18}O$ values of river water and precipitation in the urban area, and therefore did not study the changes in specific runoff flows. As described in our previous methods regarding MRT, the amplitude A of the sinusoidal exponential regression model is an important parameter in the calculation of MRT for river water, and there is some uncertainty in applying MRT to longer time scales and larger catchments. The model generally assumes certain steady-state conditions in the catchment function, which is clearly unrealistic (McGuire et al., 2005), and in order to minimize errors, in this case we weighted precipitation $\delta^{18}O$ in an attempt to characterize recharge, especially in basins with pronounced climatic seasonality (Uhlenbrook et al., 2002). Nonetheless, studies elsewhere have shown that the model remains a reliable tool for estimating MRT (Stewart and McDonnell, 1991; Soulsby et al., 2001; Asano et al., 2002; Tekleab et al., 2014; Wang et al., 2022 ), thank you for your help.

**Response:** According to your suggestion, we have redrawn Figure 1 to make the spatial distribution of sampling points clearer, thank you for your suggestion.

[Figure]

Figure 1 (a) The location of the study area, (b) Comprehensive observation system for the study area, (c) Urban surface water sampling points (from Google Maps), (d) Common urban landscape dams in SYR Basin.

5. It is probably just a formatting issue, but Figure 3 needs to move up close to Table 2 in §4.1 when it is first mentioned, as it is at the end of the next section.

**Response:** According to your comment, we have adjusted the position of Figure 2 to make it easier to read, thank you for your suggestion.

6. Figure 2 needs to move in to §3.3 where it is referenced.

**Response:** Based on your suggestion, we have adjusted the position of Figure 2 to make it easier to read, thank you for your suggestion.

7. I may be wrong, but it appears that both panels in Figure 4 are identical.

**Response:** I am very sorry for our negligence, we have modified Figure 4, it is now named Figure 6, and the modified picture is as follows.

[Figure]

Figure 6 Evaporation losses from surface water in different areas of the SYR (a) Upper reaches mountainous area, (b)Middle reaches urban areas.

8. More detail can be introduced regarding Figure 5 in particular – all we get are the regression coefficients. There is some text saying that winter has low δ18O values, but if the parameters of the regression were in a table, then the phase lag could show how well synchronised they all are except for S2, and the amplitude also might vary spatially (urban vs regional) as another item. Then there could also be more insight into why S2 was different, which is the only interesting point.

**Response:** Thank you for your comments and suggestions, we have analyzed Fig. 5 (now Fig. 7) in more detail, analyzed the reasons for the differences in the amplitude of variation ($A$) at different sampling points, and analyzed in detail the factors affecting the MRT, and the details of the modifications are as follows, thank you for your help.

Specifically:

[revised manuscript text omitted]

9. Other commenters have mentioned the issues with missing references and incomplete citations.

**Response:** We have corrected all the literature citation errors, which are indicated in red in our newly uploaded manuscript, thank you for your suggestions.

[revised manuscript text omitted]

---

## Author Comment (AC3)

**Manuscript Number: hess-2023-96**

**Revision notes:**

We would like to thank the editor and reviewers for the valuable and comments, which were very helpful for the further improvement of the manuscript. We have looked through each comment the editor and reviewers raised and responded and incorporate it in the revised manuscript. For your review, we have attached the revised PDF manuscript at the end. The following is the detailed response:

**Reviewer #1:**

The article is presenting an interesting set of data and applied some solid methodologies in order to show the impact of urban activities (broad sense).

The angle taken by the article is very interesting. However, there are more assumptions than scientific demonstration partly because the authors have long temporal series at a few SW sampling points when the scientific demonstration is more looking for spatial variability than temporal variability. The GW data is not really used. The article should be strengthened by a better use of the collected data and additional and more self-explanatory figures.

No figure is really convincing as the explanation factors remains quite "vague", land cover (which type? What variability through the basin?), soil permeability (data on that is not provided). Spatial information on these parameters is missing.

**Response:** Thank you for your valuable comments and suggestions on our manuscript. Based on your comments and suggestions, we have enhanced the analysis and discussion of the isotope data, strengthened the groundwater analysis, redescribed the spatial variability of the results, illustrated our results with more figures, and analyzed in detail the influencing factors of the MRT, with the revised sections shown in blue.

In addition, for your review, we have included a PDF of the revised manuscript at the end of this document.

**Specific comments:**

1. Line 43: the reference (Cho) is a very specific case and most probably not appropriate to be cited here.

**Response:** We agree with your suggestion and have modified the expression here. We have changed "Urbanization has led to a dramatic increase in water consumption, significantly impacting groundwater quality (Cho et al., 2009)." to "Urbanization exacerbates water depletion and has far-reaching impacts on groundwater (Flörke et al., 2018; McDonough et al., 2020)."

2. Line 62: it is not clear why these three references were cited here. There are many other studies on GW-SW studies so may be better cite local studies or global ones.

**Response:** We have modified the study cited here, we have changed "Isotopes that are stable of hydrogen and oxygen are very useful tools for investigating hydrological issues that are connected to surface water and groundwater sources (Gat,1996, Tetzlaff et al.,2007, Sanda et al.,2017)." to "Isotopes that are stable of hydrogen and oxygen are very useful tools for investigating hydrological issues that are connected to surface water and groundwater sources (Förstel and Hützen, 1983; Fekete et al., 2006; Vystavna et al., 2021)."

3. Line 132: the *river* sampling location

**Response:** We have revised the expression in response to your comments,we have changed"The sampling location" to "the river sampling location".

4. Line 135: Are all GW bodies sampled ? or a selection of GW bodies ? please specify

**Response:** We have provided a detailed description of groundwater collection, with detailed revisions below. We have changed "Samples of groundwater bodies were obtained at 7 sampling stations around the basin" to "Artesian well water was collected as groundwater samples at seven sampling locations around the basin", and is indicated in Fig. 1.

[Figure]

Figure 1 (a) The location of the study area, (b) Comprehensive observation system for the study area, (c) Urban surface water sampling points(from Google Maps), (d) Common urban landscape dams in SYR Basin.

5. Line 191: remove big cap for The

**Response:** We have modified the formulation here.

Specifically: Periodic regression analysis and the mean residence time (MRT)

6. Line 193-195: paragraph to be changed in order to highlight the method and not the specific Slovenian case

**Response:** We have revised the expression in response to your comments. We have changed "Precipitation and surface water samples were collected from a variety of locations across Slovenia,as well as from Belgrade, Serbia, for the Sava and Danube rivers. Seasonal fluctuations in $\delta^{18}O$ levels were analyzed using periodic regression analysis to determine how these levels changed over time.This method entailed fitting seasonal sine wave curves to annual $\delta^{18}O$ variations using least squares optimization (Rodgers et al.,2005)" to "Seasonal fluctuations in $\delta^{18}O$ levels were analyzed using periodic regression analysis to determine how these levels changed over time.This method entailed fitting seasonal sine wave curves to annual $\delta^{18}O$ variations using least squares optimization (Rodgers et al.,2005)."

7. Line 219: what is the seasonal variation of isotopes? Is the effect of dam similar in all season? Only average means are given and fig 3 is difficult to read as all water were represented

**Response:** I'm sorry that this sentence was not clearly expressed, which has caused you a misunderstanding. According to your suggestion, we have described the seasonal variations of isotopes, according to our results, the effect of urban landscape dams on river water and groundwater is similar, and some degree of evaporation occurs in both river water and groundwater, and we have modified Figure 3 to make it easier to understand, and the following is the detailed information of our modification. Specifically:

In both surface water and groundwater, $\delta D$ and $\delta^{18}O$ showed significant seasonal variations (Fig. 3). Seasonal variations were more pronounced in surface water than in groundwater, with surface water showing the largest amplitude in spring and the smallest amplitude in fall, while groundwater showed closer amplitudes in all seasons, which also indicates that groundwater is less disturbed.

[Figure]

Figure 3 Relationship between $\delta$D and $\delta^{18}$O in various water bodies in the SYR Basin during different seasons (a) Spring, (b) Summer, (c) Autumn, (d) The contrast between RWL, GWL, LMWL and GMWL throughout the sampling period.

8. Fig2: Where the GW points were taken comparing to the distance to surface water and dam? No location is given in the paper. This is also most probably the reason why GW variability is not discussed in the chapter 4.1

**Response:** In response to your suggestion that we also increase the distance to dams near groundwater, we have modified Figure 2 and added some groundwater isotope data to make it easier to understand. The revised Figure 2 is shown below.

[Figure]

Figure 2 Longitudinal variation of $\delta$D and $\delta^{18}$O in river water and groundwater in the SYR Basin.

9. Line 249: Global meteoric

**Response:** Due to our negligence, this error was caused, and we have revised the content, We have changed "global geteoric water line" to "global meteoric water line".

10. Fig3: very small figures and difficult to read. Not all explanation in the text can be confirmed by the figure due to too many information in very small size figures.

**Response:** According to your suggestion, we have modified Fig 3. The detailed revisions are as follows.

[Figure]

Figure 3 Relationship between $\delta D$ and $\delta^{18}O$ in various water bodies in the SYR Basin during different seasons (a) Spring, (b) Summer, (c) Autumn, (d) The contrast between RWL,GWL,LMWL and GMWL throughout the sampling period.

11. Line 252-255: the sampling points are not so close to the LML, some are even quite far. Also possible GW discharge to the surface water and possible recharge mechanism looks possible for the GW. More discussion would be needed

**Response:** Yes, there are some surface water sampling sites far from the LMWL, and after we rechecked the data, we eliminated some of the isotopic anomalies that may have been caused by experimental and other factors, and re-plotted Figure 3.In case the response is too cumbersome and inconvenient to view and read, the revisions to Figure 3 are described in Response 10 above, and I thank you for your help. In addition, we have described in detail the interactions between surface water and groundwater and provided detailed information on the revisions.

Specifically:

We compared monthly variations in isotopic values of groundwater near the city with monthly variations in river water from a landscaped dam and found that the monthly variations in groundwater near the city were closely related to river water from a landscaped dam. The concentration of groundwater sampling sites near the city near the sampling sites of the dam water indicates that the groundwater around the city has similar isotopic signatures to the dam and river water. This suggests that groundwater near the city is recharged by river water during the summer months. In addition, we demonstrated this by comparing the data of the dam river water with the groundwater level. In addition, a portion of the groundwater sampling sites around the city are located in the lower right corner of the LMWL, which suggests that the groundwater around the city may also experience some degree of evaporation.

[Figure]

Figure 4 (a) Relationship between $\delta^{18}$O and $\delta$D of groundwater around city and urban river water; (b) Monthly variations of $\delta^{18}$O in groundwater around city; (c) Monthly variations of $\delta$D in groundwater around city.

In addition, we also compared and analyzed the changes of groundwater isotope values with those of groundwater around the city in the whole basin, and found that there was a close correlation between the changes of groundwater around the city and those of the river, while the other groundwater isotope values did not have a strong correlation with the river. In the urban area, the mean values of $\delta D$ and $\delta^{18}O$ of the dammed river water were -8.26‰ and -49.88‰, respectively, while the mean values of $\delta D$ and $\delta^{18}O$ of the groundwater around the city were -8.44‰ and -50.36‰, respectively, which indicated that the $\delta D$ and $\delta^{18}O$ values of the groundwater around the city were similar to those of the river water in the dammed city. In addition, the isotopic mean values of $\delta D$ and $\delta^{18}O$ of groundwater throughout the SYR basin were -8.73‰ and -54.78‰, which are significantly different from the isotopic values of river water in the urban dams.

[Figure]

Figure 5 (a) Monthly variations of $\delta^{18}O$ in groundwater around city, (b) Monthly variations of $\delta D$ in groundwater around city.

12. Line 257: it is not because of the different correlation line that we can conclude in a variation of IC from up to down.

**Response:** I'm sorry that this sentence was not clearly expressed, which has caused you a misunderstanding. we have revised the narrative, and here is the detailed revision information.

We have changed "The SYR Basin surface water samples that were collected exhibited a linear regression of $\delta D=5.63\delta^{18}O-6.11$, which revealed a spatial variation in isotopic composition from upstream to downstream. Where the river water line (RWL) intercept and slope show a trend that is first decreasing and then increasing as one moves from upstream to downstream. This demonstrated the presence of significant isotopic differences in the water." to "Overall, the H-O isotopic composition of surface water samples from the SYR showed a linear regression of $\delta$D = 5.63$\delta^{18}$O - 6.11, and the slope of RWL was the largest in the autumn (slope = 6.65) and the smallest in the summer (slope = 5.56), which indicated that the river water evaporated the weakest in the autumn and the strongest in the summer."

13. Line 258-261: Sentence not clear

**Response:** We apologize for the reading difficulty caused by the lack of clarity of presentation here. We have revised the sentence and the changes are listed in point 12 of the response, thank you for your help.

14. Line 262-265: location of GW points and variation in comportment is missing here. It looks that there is a large IC variations between one GW point to another

**Response:** Based on your comments, we have revised the presentation.

Specifically:

And it is interesting to note that groundwater also shows significant enrichment near the urban landscape dams (Fig. 2), indicating that groundwater is also affected by evapotranspiration, mainly because the Wuwei urban area is in the region of a large alluvial fan in front of the mountains, the sand and gravel aquifers are very permeable, and the depth of groundwater burial is shallow, making the groundwater more susceptible to the effects of evaporation.

15. Line 267-271: this is theoretical but does not give much information to the reader

**Response:** Based on your suggestion, we have decided to remove this erroneous and confusing statement.

16. Line 285 and fig4: is it evaporation loss calculated for each points for the whole period? Is a) and b) environment explained somewhere in the text? Which SW point in each category ?

**Response:** Yes, we calculated evaporation losses for each surface water sampling site in the upstream and midstream for the entire sampling period.For the a) and b) environments, we made relevant interpretations, and detailed modification information is provided below.

Specifically:

Differences contributing to evaporation losses from the river in the upstream and midstream urban areas can be explained mainly by the landscape characteristics of the basin. In the upstream of the Shiyang River, higher vegetation cover and atmospheric humidity in the mountainous areas result in weaker evaporation losses, while the midstream are dominated by urban land, and urban landscapes increase the watershed area and slow down the river, exacerbating evaporation losses losses from the river.

17. Line 308: assumptions, not demonstrated, a ref would be needed

**Response:** Based on your suggestion, we have added the relevant literature here, and the following is the detailed revision information.

Specifically:

We have changed "The $\delta^{18}O$ levels of precipitation reported in the SYR Basin have an excellent regularity ($R^2=0.46$) and a seasonal patterns trend that effectively depicts the nfluence of the monsoon climate on the local environment" to "The $\delta^{18}O$ levels of precipitation reported in the SYR Basin have an excellent regularity ($R^2=0.46$) and a seasonal patterns trend that effectively depicts the nfluence of the monsoon climate on the local environment (Zhu et al., 2019)."

**Response:** According to your suggestion, we have analyzed the factors affecting MRT specifically, about the uncertainty of MRT calculation we have mentioned in the previous 18 points, thank you for your help below is the specific modification information.

Specifically:

The Dunnett's test revealed a significant difference ($P < 0.05$) between the MRT of the river and the annual magnitude of $\delta^{18}O$ of the river. We further investigated the relationship between the estimated mean residence time and basin landscape features such as topography (Fig. 8). Using the digital elevation model (DEM) to calculate the mean slope of the SYR basin, we found that the mean residence time was also strongly correlated with the mean basin slope ($R^2 = 0.63$), and that the upper reaches of the Shiyang River basin are mainly high-elevation mountainous areas, where the topography is sloped, but where the vegetation cover is high and dominated by alpine meadows, subalpine scrub and Qinghai spruce (Zhang et al. 2023), the greater slope leads to a higher gravitational potential, which tends to result in a negative correlation with mean residence time (McGuire et al., 2005), which also contributes to the potentially higher MRT values in the upstream mountains. In our study, catchment area (CA) had a low correlation with MRT ($R^2 = 0.40$), and a weak relationship between catchment area and MRT has been observed in other studies (McGlynn et al., 2003; McGuire et al., 2005).

[Figure]

Figure 8 Correlation between mean slope of the basin (a), catchment area (b), sand clay loam ratio (c), ratio of residential and industrial areas to total basin area (d) and MRT.

Soil is an important component of basin hydrology, and the physical properties of soil, such as water-holding capacity and pore space distribution, have an important influence on the response to precipitation in the basin, and the sand-clay-loam soil ratio is used here to investigate the possible relationship with MRT. The results showed that the content of sand clay loam ratio showed a strong positive correlation with MRT ($R^2$=0.62). Wuwei City is located in the pre-mountain flood-fan belt, and the soil is dominated by sandy soil (Zhang et al., 2023), which is loose in texture, has good permeability and good water retention properties, and is mainly used for agricultural cultivation. Its good permeability increases the vertical movement of water and the length of flow paths, leading to a longer MRT. There is a strong negative correlation between the MRT and the ratios of resident and industrial areas (RI), which also indicates that as urbanization progresses, with the increase of urban land, this undoubtedly leads to a significant shortening of the MRT. However, the MRT in the mid-river urban area is not much shorter as compared to the downstream, which may be attributed to the fact that the mid-river The large number of landscape dams constructed in the urban areas, currently 51 urban landscape dams have been built in the peri-urban areas of Wuwei City, and the considerable number of landscape dams may have counteracted the impact of the urban land use, resulting in a lengthening of the MRT in the middle reaches as well.

**Response:** In response to your suggestions, we have revised the presentation of this section, and here are the details of the revisions.

Specifically:

In terms of the impact on runoff, it is mainly reflected in the increase of surface impermeability due to urbanization, the land use area of Wuwei urban land increased by about 134.38 km2 from 2010 to 2018, which greatly weakened the infiltration process in urban areas, and the rainfall runoff process simulated by sinusoidal cyclic regression method showed that there were significant differences in the river metro in different parts of the Shiyang River Basin, and that the middle reaches of the river had the highest degree of urbanization, and the time of the metro was the shortest, which further increases the contribution of rainfall to runoff. Regarding the effect of urbanization on evapotranspiration, a large number of dams were constructed on the Shiyang River and flowed through the urban area of Wuwei, causing significant evapotranspiration losses, in addition, these landscape dams also led to hydrogen and oxygen isotope enrichment (Fig. 10), and the numerous reservoirs that were constructed for the construction and development of the city (Ma et al., 2010), and these reservoirs also contributed to a significant evapotranspiration losses effect, which has been previously confirmed in our study was also confirmed (Sang et al., 2023). On the other hand, our study found that the isotopic compositions and trends of urban nearshore groundwater were similar to those of surface water, which suggests that there is a close correlation between urban nearshore groundwater and river water, and that the difference in water levels between river water and groundwater may be the main reason for river recharge of urban nearshore groundwater (Fig. 4). In the rainy season, the river level gradually rises, which decreases the difference between the water levels of urban nearshore groundwater and river water, and the river water recharges the groundwater, and in the dry season, the river level decreases, and the urban nearshore groundwater, which is buried at a shallow depth, in turn recharges the river.

In addition, the growth of urbanization has had a dramatic impact on the water environment in cities, where water problems occur frequently (Giri and Qiu, 2016; Ma et al., 2022). Urbanization has increased impervious surfaces such as parking lots, rooftops, roads, and sidewalks, leading to increased runoff, which creates additional pathways for pollutants to be transported from landscapes to water bodies ((Ren et al., 2014; Wilson and Weng, 2020; Nolan et al., 2023). On the other hand, agricultural activities have increased some of the fertilizers, pesticides, herbicides and dairy manure in the farmland into the nearest water bodies, which can directly and indirectly affect will reduce water quality (Yu et al., 2020). The Shiyang River Basin in the Northwest Arid Zone is an inland river basin with the highest development intensity and the sharpest conflict between water supply and demand in the region. The Liangzhou district in the central part of the Shiyang River basin is the most densely populated artificial oasis with the largest scale of water demand in the entire basin. Our previous study found that direct discharge of industrial and community domestic wastewater into the river led to deterioration of surface water quality around the Shiyang River basin (Ma et al., 2021). In addition agricultural activities have less impact on the upper reaches of the Shiyang River and relatively more impact on the middle and lower reaches , and the application of nitrogen-based fertilizers during agricultural cultivation is the main cause of high $NH_4^+$ and $NO_3^-$ concentrations in the area (Ma et al., 2021), which may also lead to increased salinity and accelerated eutrophication of the river, threatening the safety of the basin's water environment. Overall, human activities (urbanization) may alter the water cycle processes inherent in inland river basins, and the implications of such changes need to be further explored.

26. Line 425-439: these are processes seen with data – OK

**Response:** I'm sorry that this sentence was not clearly expressed, which has caused you a misunderstanding. We have revised the presentation of this section and the details of the revisions are provided below.

Specifically:

In this study, we investigated the hydrometeorological and isotopic data of the Shiyang River Basin from 2017 to 2021, and our investigations showed that urbanization had a significant impact on the water cycle of the basin.The results showed that the isotopic values of the river water showed a significant enrichment from upstream to downstream, but facilities such as landscape dams and reservoirs in the urban area significantly altered this natural pattern, and the isotopic values of the river water in the urban area ($\delta$D=-48.31‰; $\delta^{18}$O=-7.49‰) were higher than those of the natural river water ($\delta$D=-55.77‰; $\delta^{18}$O=-8.98‰), and landscape dams aggravated the evaporation losses of river water, due to the increase of urban land area, which accelerated the rainfall-runoff conversion process, the residence time of surface water in different regions of the Shiyang River Basin had obvious differences, and the MRT from the upstream to the downstream showed a fluctuating downward process, which was shortened from 1,126 days in the upper reaches to 941 days in the lower reaches, and the MRT was mainly controlled by the basin's landscape features. In addition, there was a strong relationship between the isotopic composition of the reservoir and the surrounding groundwater. Overall, urbanization has a profound impact on the hydrological system of the basin, and the results of this study can provide some references for future research on urbanization and the water cycle, and improve our understanding of the hydrological processes of basins in arid zones.

27. Line 440-443: not really demonstrated but could be done with the available data most probably

**Response:** Based on your suggestions, we have modified the presentation here, and the modified information has been responded to in question 26 above, thank you for your help.

28. Line 443-445: assumption only

**Response:** Based on your suggestions, we have modified the presentation here, and the modified information has been responded to in question 26 above, thank you for your help.

29. Many references are not properly cited in the text :

**Response:** I am very sorry for our negligence. We have re-cited the references, thank you for your help.

30. Line 508 : small caps

**Response:** Based on your suggestions, we have modified the references here.

31. Line 521 : small caps

**Response:** Based on your suggestions, we have modified the references here.

**Response:** Based on your suggestions, we are citing the research results updated here.

Reference:

[revised manuscript text omitted]

---

## Author Response (AR2)

**Manuscript Number: hess-2023-96**

**Revision notes:**

We would like to thank the editor and reviewers for the valuable and comments, which were very helpful for the further improvement of the manuscript. We have looked through each comment the editor and reviewers raised and responded and incorporate it in the revised manuscript. For your review, we have attached the revised PDF manuscript at the end. The following is the detailed response:

**Reviewer #2:**

hess-2023-96 "Effects of urbanization on the water cycle in the Shiyang River Basin: Based on stable isotope method" Rui Li, Guofeng Zhu, Siyu Lu, Liyuan Sang, Gaojia Meng, Longhu Chen, Yinying Jiao, Qinqin Wang

**1. The organisation of the paper, particularly section and sub-section numbering, is an issue. The title of "Section 2. Systems, Data, and Methods of Observation" implies many things yet we have just 2 paragraphs of site description and Figure 1, which should be "S2.1 Site Description". The first occurrence of Section 3 (p7 line 139), that should be S2.2, lists the data sample names and an equation for isotope difference, while the second occurrence of Section 3 (p9 line 168), that should be S2.3, lists the actual analyses to be performed on the isotope measurements. There is something to be said for the old-fashioned format of 1-Introduction, 2-Methods and Materials, 3-Results, 4-Discussion and 5-Conclusion.**

**Response:** We have changed the numbering of the chapters based on your comments, here are the details of the changes:

Introduction

Observation Systems and Data

   2.1 Study Area

   2.2 Sampling and data analysis

Methods

   3.1 Calculation and indication of *d-excess*

**2. The way the data is presented it seems that S1 to S10 are sequential and cascading along the same reach which does not appear to be the case. Looking at Table 2, the obvious outlier is S6, which appears to be on a south-eastern tributary of the main channel. If this is not downstream of the urban dams, and therefore may not be affected by their isotopic concentration, its inclusion in the overall sequence is misleading. In fact, looking at all the sites in Figure 1, S1 and S2 are on a different tributary to S3, S4 and S5 in the urban dam area, and different again to S6. This leads to only S7 to S10 being sequential on the same channel reach, with the major reservoir between S9 and S10. What you have are three groups: (1) unimpaired river water (S1, S2, S6), (2) urban dam affected river water (S3, S4, S5), and (3) downstream mixed river water (S7, S8, S9, S10) with S10 also affected by the major reservoir.**

Response: We have explained the key issue you mentioned. The detailed explanation is as follows.

Due to our drawing error, Figure 1 is presented in an incorrect form. We have modified Figure 1. The specific modification information is as follows.

[Figure]

Figure 1 (a) The location of the study area, (b) Comprehensive observation system for the study area, (c) Urban surface water sampling points (from Google Maps), (d) Common urban landscape dams in SYR Basin.

The Shiyang River Basin is an inland river basin in the arid region of Central Asia. It is mainly composed of eight rivers from east to west, including the Dajing River, Gulang River, Huangyang River, Zamu River, Jinta River, and Xiying River. In addition to the Dajing River, six rivers in the central part merge into the Shiyang River near Wuwei, flow into the Hongyashan Reservoir, and then enter the Minqin Basin.

We chose the Xiying River Basin located in its upper reaches as the upstream runoff area. Since the Xiying River Basin is one of the areas with the richest rainfall in the Shiyang River Basin, and there are glaciers at its source, the Xiying River Basin has become a typical runoff-producing area in the Shiyang River Basin, so two sampling points S1 and S1 were set up. . The three sampling points S2, S3, S4 and S5 are located in the urban landscape dam area and are most obviously affected by urbanization. The value is mainly affected by the tributaries, resulting in a more depleted S6 isotope value. Moreover, S6 is far away from the urban area and is basically not affected by the river water in the upstream city. S7, S8, S9, and S10 are all located in the downstream mainstream, while s10 is located in the Hongyashan Reservoir, which also results in a longer residence time. Therefore, S10 also represents the isotope composition characteristics of the reservoir water. After S10, the Shiyang River is mainly transported downstream in the form of main water pipes.

At the watershed scale, S1-S10 are located on continuous rivers whose connectivity is changed by reservoirs, dams and other water facilities, which is a result of urbanization and is one of the main causes of changes in the water cycle of the watershed and is our main study content.

Referencce:

Sang, L., Zhu, G., Qiu, D., Zhang, Z., Liu, Y., Zhao, K., Wang, L., and Sun, Z.: Spatial variability of runoff recharge sources and influence mechanisms in an arid mountain flow-producing zone, Hydrological Processes, 36, https://doi.org/10.1002/hyp.14642, 2022.

Sang, L., Zhu, G., Xu, Y., Sun, Z., Zhang, Z., and Tong, H.: Effects of Agricultural Large-And Medium-Sized Reservoirs on Hydrologic Processes in the Arid Shiyang River Basin, Northwest China, Water Resources Research, 59, e2022WR033519, https://doi.org/10.1029/2022WR033519, 2023.

Zhang, G., Su, X., Ayantobo, O. O., Feng, K., and Guo, J.: Remote-sensing precipitation and temperature evaluation using soil and water assessment tool with multiobjective calibration in the Shiyang River Basin, Northwest China, Journal of Hydrology, 590, 125416, https://doi.org/10.1016/j.jhydrol.2020.125416, 2020.

Zhu, G., Sang, L., Zhang, Z., Sun, Z., Ma, H., Liu, Y., Zhao, K., Wang, L., and Guo, H.: Impact of landscape dams on river water cycle in urban and peri-urban areas in the Shiyang River Basin: Evidence obtained from hydrogen and oxygen isotopes, Journal of Hydrology, 602, 126779, https://doi.org/10.1016/j.jhydrol.2021.126779, 2021.

**3. Figure 3 has three panels that are made up of three smaller graphs, and all panels need to be labelled to fit with the caption. The two panels in Figure 6 need the same vertical scale, so that the data are visually different.**

**Response:** Due to our negligence that caused this error, based on your suggestion, we have modified Figures 3 and 6 to make them easier for the reader to understand, and the following is the specific change information.

[Figure]

Figure 3 Relationship between $\delta$D and $\delta^{18}$O in various water bodies in the SYR Basin during different seasons (a,d,g represent spring, summer, autumn; j represent the comparison of RWL, GWL, LMWL and GMWL during the entire sampling period;.b-c, e-f, h-i represent the distribution of $\delta$D and $\delta^{18}$O in river water and groundwater in spring, summer and autumn).

[Figure]

Figure 6 Evaporation losses from surface water in different areas of the SYR (a) Upper reaches mountainous area, (b) Middle reaches urban areas.

**4. The caption for Figure 8 needs to be worded in a more standard way:**

**"Correlation between MRT and (a) mean slope of the basin, (b) catchment area, (c) sand clay loam ratio, and (d) percentage area of residential and industrial use in the basin, with 95% Confidence and Prediction bands."**

**Response:** In response to your suggestion, we have modified the title of Figure 8 as follows:

[Figure]

Figure 8 Correlation between MRT and (a) mean slope of the basin, (b) catchment area, (c) sand clay loam ratio, and (d) percentage area of residential and industrial use in the basin, with 95% Confidence and Prediction bands.

**5. Figure 10 does not add to the text provided, and can be omitted.**

**Response:** Based on your suggestion, we have removed Figure 10, thank you for your help.

[revised manuscript text omitted]